# Influences of Semaphorin 3A Expression on Clinicopathological Features, Human Papillomavirus Status, and Prognosis in Oropharyngeal Carcinoma

**DOI:** 10.3390/microorganisms8091286

**Published:** 2020-08-22

**Authors:** Hai Thanh Pham, Satoru Kondo, Kazuhira Endo, Naohiro Wakisaka, Yoshitaka Aoki, Yosuke Nakanishi, Kina Kase, Harue Mizokami, Makoto Kano, Takayoshi Ueno, Miyako Hatano, Makiko Moriyama-Kita, Hisashi Sugimoto, Tomokazu Yoshizaki

**Affiliations:** Division of Otolaryngology-Head and Neck Surgery, Graduate School of Medical Science, Kanazawa University, Kanazawa, Ishikawa 920-8640, Japan; thanhhai.hpmu@gmail.com (H.T.P.); endok@med.kanazawa-u.ac.jp (K.E.); wakisaka@med.kanazawa-u.ac.jp (N.W.); ltd.express.0755@med.kanazawa-u.ac.jp (Y.A.); nakanish@med.kanazawa-u.ac.jp (Y.N.); kina@med.kanazawa-u.ac.jp (K.K.); haruesun@med.kanazawa-u.ac.jp (H.M.); makoto-kano@med.kanazawa-u.ac.jp (M.K.); uenotaka@med.kanazawa-u.ac.jp (T.U.); mhatano@med.kanazawa-u.ac.jp (M.H.); mkita@med.kanazawa-u.ac.jp (M.M.-K.); sugimohi@med.kanazawa-u.ac.jp (H.S.); tomoy@med.kanazawa-u.ac.jp (T.Y.)

**Keywords:** SEMA3A, HPV, oropharyngeal carcinoma

## Abstract

Human papillomavirus (HPV) infection is now identified as a major etiologic factor for oropharyngeal cancer (OPC), and HPV positivity is well established better prognostic marker in OPC. Now, predictable markers for the prognosis of the patients who are stratified by HPV has been investigated in. Semaphorin 3A (SEMA3A) is a well-known axon guidance molecule in the nervous system. It is also known as a tumor suppressor in various cancers. In the present study, we examined the relationships between SEMA3A and clinicopathologic features, especially HPV status, and neoangiogenesis, and its prognostic significance for OPC patients. Thirty-two OPC patients and 17 normal patients were analyzed for SEMA3A expression by immunohistochemical analysis. We also analyzed 22 OPC specimens for CD34 expression as a marker of neoangiogenesis. SEMA3A was significantly downregulated in OPC compared with chronic tonsillitis tissues (*p* = 0.005). SEMA3A expression was negatively correlated with CD34 expression (*r* = −0.466, *p* = 0.033). Moreover, the higher SEMA3A expression cohort showed better survival than the lower SEMA3A expression cohort regardless of HPV status (*p* = 0.035). These results suggest that SEMA3A expression is a prognostic marker for survival regardless of HPV status and is associated with anti-angiogenesis in OPC.

## 1. Introduction

Head and neck cancer (HNC) is the sixth most prevalent cancer worldwide and is considered the sixth leading cause of cancer mortality [1]. Oropharyngeal cancer (OPC) has become the common subsite of HNC, oral sex has been believed to be involved in the development of Human papillomavirus (HPV)-induced OPC [2]. OPC incidence has increased over the last 20 years in several countries [3,4,5,6], including the United States [7], Canada [8], and Japan [9]. There were more than 92,000 new cases of OPC and 51,000 associated deaths worldwide in 2018 [10]. Nearly all cases of cervical cancers can be attributable to HPV infection, in contrast, OPC has two distinct etiologies: tobacco and alcohol consumption, and HPV infection [11]. Making this distinction is clinically important because HPV (+) OPC has a more favorable prognosis compared with OPC that is HPV(−) [12,13,14,15,16]. HPV (−) OPC tends to affect older patients with history of intensive tobacco smoking and alcohol drinking, in contrast, patients with HPV (+) OPC are more likely to be younger without extensive tobacco and alcohol use [17]. Today, it is urgently important to identify prognostic factors in the OPC patients stratified by HPV status.

Semaphorins are a large family of axon guidance. Semaphorins include both secreted and membrane-bound proteins that were initially implicated in the development of the nervous system and axon guidance. Recently, Semaphorin 3A (SEMA3A) has been shown to regulates cell adhesion, cell motility, angiogenesis, immune responses, and tumor progression [18,19]. In the mammalian system, 20 semaphorins have been grouped into five classes (semaphorins 3–7), in which class 3 semaphorins are secreted proteins and classes 4–6 semaphorins are transmembrane proteins. SEMA3A is identified as a potent tumor suppressor in some cancers (breast cancer and prostate cancer [20,21]), inhibits endothelial cell adhesion and migration [22], induces the collapse of the actin cytoskeleton and apoptosis, reduces angiogenesis in vitro [23]. The expression levels of SEMA3A are also overexpressed in meningioma tissues, and the expression level is negatively correlated with the microvessel density (MVD) of the tumor [24]. MVD has also been determined to be an effective prognostic factor and is related to survival in OPC [25].

However, until now, the effect of SEMA3A in OPC had not been studied intensively and remained unclear. Previously, there were some prognosis factors that have been shown to have mutually connections with HPV in regard to OPC pathology. Some antiviral factors, such as members of the apolipoprotein B mRNA-editing catalytic polypeptide 3 (APOBEC3, A3) family, which are inducible by estrogen, were found to induce hypermutation of HPV DNA in vitro [26]. A3s also was found to contribute to the improved patient responses in HPV infected patients with head and neck cancer by activating BER (base excision repair) in HNSC (head and neck squamous cell carcinomas), mediating repair of cisplatin ICLs (interstrand crosslinks) [27]. A3A (APOBEC3A) expression may be upregulated by HPV16 infection, thereby increasing the chance of viral DNA integration [26]. Furthermore, some research findings indicate that there was a close association among ERα (estrogen receptor α), A3A, and survival in HPV (+) OPC. ERα also has been demonstrated to be a favorable prognostic factor for survival in HPV (+) OPC [28].

Therefore, we focused on the expression of SEMA3A. Finally, the aim of this study is to elucidate expression of SEMA3A may associate with prognosis in OPC patients. Furthermore, we focus on its relationships with CD34 expression and HPV status.

## 2. Materials and Methods

### 2.1. Patients and Tissue Samples

The study was a retrospective cohort study. From June 2004 to February 2015, the biopsy specimens of 32 OPC patients who underwent biopsy and 17 normal patients who underwent tonsillectomy were obtained from the Division of Otolaryngology-Head and Neck Surgery, Kanazawa University Hospital. Thirty-two cancer biopsy specimens were obtained from OPC patients and then verified by pathological sectioning. None of these patients had received radiotherapy or chemotherapy prior to biopsy. Seventeen noncancerous tonsil tissues were obtained from normal patients who had chronic tonsillitis. All tissues were obtained with the consent of patients. Clinical information related to 32 cancer patients—including age, sex, and TNM stage—was also obtained. The TNM stage was defined by the guidelines of the UICC, eighth edition, 2018. Distant metastasis was determined by radiological examination. Overall survival, which was defined as the time from diagnosis to the time of patient death, date of censoring due to loss of follow-up or last follow-up, was used as a measure of prognosis and ranged from 2 to 121 months. This study was approved by the Ethics Committee of Kanazawa University, (IRB #2012-031).

### 2.2. Immunohistochemistry

Among the 32 OPC biopsy specimens, all 32 were immunohistochemically examined for SEMA3A, and 21 were examined for CD34. Tumor angiogenesis is an important step in tumor growth and metastasis. Antibodies against CD34, which are specific to the vein endothelium, were used for MVD evaluation [25]. All specimens were fixed in a 10% formaldehyde solution and embedded in paraffin. Tissues were deparaffinized in xylene and rehydrated through an alcohol gradient. Endogenous peroxidase activity was blocked with 3% hydrogen peroxide for 10 min after three washes in phosphate-buffered saline at pH 7.2. The slides were boiled in 10 mM sodium citrate retrieval buffer (pH 6.0) for 20 min for antigen retrieval. After rinsing with PBS, the tissue sections were incubated with a protein block (Dako, Glostrup, Denmark) for 20 min and incubated overnight at 4 °C with the following antibodies as the primary antibody: rabbit anti-SEMA3A (1:200, rabbit polyclonal to SEMA3A, Abcam, Cambridge, UK) and rabbit anti-CD34 (1:100, rabbit monoclonal to CD34, Abcam, Cambridge, UK). The sections were washed three times with PBS. Next, the sections were exposed to EnVision + secondary antibody (Dako) for 30 min. The reaction products were developed by immersing the sections in 3′,3-diaminobenzidine tetrahydrochloride solution. Subsequently, the sections were counterstained with hematoxylin. The stained sections were independently evaluated by two investigators (H.P. and S.K.) who were blinded to the clinical data using an IX83 microscope (Olympus, Tokyo, Japan).

### 2.3. Evaluation of SEMA3A Immunoreactivity

Each slide was observed by scanning the whole section at medium (×40) and high (×200) magnification under a light microscope. The immunoreactive score was defined as the proportion score multiplied by the intensity score. The criteria were as follows: (i) The intensity of SEMA3A staining was scored as 0 (negative), 1 (weak), 2 (medium), or 3 (strong); (ii) the percentage of positive-staining cancer cells in tumor specimens or positive-staining epithelial cells in the specimens was scored as 0 (0–5%), 1 (6–25%), 2 (26–50%), 3 (51–75%), or 4 (76–100%). The total score ranged from 0 to 12. The expression of SEMA3A was divided into the following groups: Negative immunoreactivity was defined as a total score of 0, a score of 1–4 was low expression, and a score of >4 was high expression.

### 2.4. Evaluation of CD34 Immunoreactivity

The vascular objects were counted to determine vascular density (vessel/mm²). Ten areas of densely concentrated microvessels (hotspots) were located using 100× magnification (objective lens × 10 and ocular lens × 10). In each case, these hotspot areas were used for counting microvessels at 400× magnification (40× objective lens and 10× ocular lens). A vascular unit was identified according to the criteria established by Weidner [29], who described a vascular unit as a cell or group of endothelial cells of a brownish color clearly separated from adjacent microvessels, tumor cells, and other connective tissue. The total number of vessels obtained in each case was the result of the total sum of vessels counted in each of the 10 microscopic fields evaluated at 400× magnification. MVD was defined as the average number of microvessels per field, which was calculated from the total number of microvessels in 10 fields [30].

### 2.5. Statistical Analyses

SPSS statistics package version 19 (IBM, New York, NY, USA) was used for data analysis. The clinical characteristics of the patients were analyzed using Fisher’s exact test or the Chi-square test. The Spearman test was used in the correlation analysis. Overall survival curve was obtained using the Kaplan–Meier method and calculated using log-rank test. *p* values less than 0.05 were considered statistically significant.

## 3. Results

### 3.1. Immunohistochemical Analysis of SEMA3A Expression in OPC Tissues

Our study included 32 patients with OPC and 17 patients with chronic tonsillitis for immunohistochemical analysis. SEMA3A was detected primarily in the cytoplasm of normal cells of tonsillitis (Figure 1A) and OPC cell (Figure 1B: Low level of SEMA3A expression in OPC tissues, Figure 1C: High level of SEMA3A expression in OPC tissues). Only 3 of 17 (17.64%) chronic tonsillitis tissues showed low SEMA3A expression, whereas 19 of 32 (59.37%) specimens of OPC tissue displayed low SEMA3A expression, indicating that SEMA3A expression was significantly upregulated in the OPC tissues compared with chronic tonsillitis tissues (*p* = 0.005) (Table 1).

### 3.2. Association between the Expression of SEMA3A and Various Clinicopathological Features

Next, we examined whether there are any associations between the expression of SEMA3A and various clinicopathological features. However, there was no significant difference in SEMA3A between the groups classified by age, sex, subsite, TNM stage, tumor stage, lymph node metastasis, HPV status, smoking history, and alcohol consumption by the Chi-square test (Table 2).

### 3.3. Association between the Expression of SEMA3A and Survival

We next assessed the relationship of SEMA3A expression with prognosis in OPC patients. Overall survival (OS) analysis was performed in the 32 patients, and the five-year OS rate was 66.5% (Figure 2A). The five-year OS rate was 44.3% for patients with low SEMA3A expression and 83.3% for patients with high SEMA3A expression, which was a significant difference (Figure 2B). Finally, we proved that the high SEMA3A expression group had a significantly longer survival than the low expression group. Regarding HPV infection, we also found that patients with low levels of SEMA3A expression had a worse prognosis than patients with high levels of SEMA3A regardless of HPV status (Figure 2C,D). The results were obtained similar to disease free survival (DFS) analysis (Appendix A). However, Cox regression analysis with OS and DFS did not show any associations of age, sex, subsite, TNM stage, tumor stage, lymph node status, HPV status, histological grade, drink, and smoking (Appendix A).

### 3.4. Association between SEMA3A Expression and MVD

We next examined whether there was any correlation between the expression of SEMA3A and CD34 in OPC tissues using IHC analysis. IHC with CD34 antibody was performed on 21 OPC specimens (Figure 3A). We revealed that the expression score of SEMA3A and the number of microvessels (CD34) in OPC samples were negatively correlated (Figure 3B). These results support the hypothesis that SEMA3A acts as a tumor suppressor or angiogenesis inhibitor.

## 4. Discussion

The present study focused on the role of SEMA3A in the prognosis and pathological angiogenesis of OPC, and its relationship with HPV status. HPV infection has been increasingly recognized as an important etiological factor for a subset of HNCs, including OPC [31]. HPV (−) OPC carries a poorer prognosis compared with HPV (+) tumors [13]. The improved survival of patients with HPV (+) tumors can be attributed in part to their remarkable treatment sensitivity, as HPV (-) tumors have worse response to chemotherapy and radiation than HPV (+) tumors [14]. HPV (+) OPC are clinically and molecularly different from HPV (−) OPC. Typically, patients with HPV (+) OPC are diagnosed with advanced- stage disease but they have better response to treatment [14]. About 68% reduced risk of death among patients with OPC who were positive for E6 antibodies versus those who were negative for E6 antibodies after adjustment for smoking was reported recently [32]. Interestingly, patients with HPV (+) tumors are shown to be less likely to have p53 mutations. The absence of p53 mutations in the tumor cell may allow an enhanced response to radiotherapy [33]. Our previous report revealed the relationship between the presence of HPV and some potential prognostic markers, such as ERα and A3A [28]. Numerous C-to-T and G-to-A hypermutations in the HPV16 genome in oropharyngeal cancer (OPC) biopsy samples suggests that HPV16 infection may upregulate A3A expression. Interestingly, A3s were known to abundantly express in HPV16 (+) OPC than in HPV (−) [26]. Higher expression of A3A mRNA and protein have been found in ERα-positive cases compared to ERα-negative ones, among HPV (+) biopsy samples [28].

SEMA3A has been identified as a candidate tumor suppressor and it is often found to be downregulated in different types of cancer, such as prostate cancer, breast cancer, and glioma. Increased SEMA3A expression was associated with better prognosis in patients with cancer, including epithelial ovarian carcinoma, gastric cancer, tongue cancer, and HNC [34,35,36,37]. A reduction in SEMA3A expression was observed in our OPC samples compared with chronic tonsillitis tissues. Moreover, overexpression of SEMA3A inhibits gastric cancer cell proliferation and migration in vitro [35]. The data presented here suggest that SEMA3A has a tumor suppressor function in OPC because our Kaplan–Meier survival analysis showed that low SEMA3A expression significantly correlated with shorter survival time in OPC patients. This result is consistent with earlier reports indicating that SEMA3A acts as an anti-tumorigenesis agent. The presence of HPV and high expression of SEMA3A are favorable prognosis factors with regard to survival in OPC, but in the present study, we could not defined a mutual relation between SEMA3A and HPV status since we found that low SEMA3A expression significantly correlated with decreased survival in OPC patients with or without HPV.

We next immunohistochemically analyzed SEMA3A expression and CD34 expression in the same tumor to evaluate the correlation of SEMA3A with the MVD. We found a significant association between high expression of SEMA3A and low MVD in the tumors. Tumor angiogenesis induces increased tumor cell circulation and it is an essential component of the metastatic pathway. Important evidence about the relationship between angiogenesis and metastasis is that tumor MVD leads to increased metastatic potential and poor survival in nearly all types of cancer [38]. Angiogenesis is a cancer hallmark, important for metastatic tumors, and is essential for the growth of lung micrometastases [38]. CD34 antibody is a credible marker for MVD evaluation. Recent studies have found associations between MVD and poor disease course in breast, lung, stomach, and bladder cancer [39,40,41].

Our results suggest that SEMA3A may be an antiangiogenic factor in OPC. The molecular mechanisms underlying the anti-tumor effects of SEMA3A are being extensively studied, the antiangiogenic factor SEMA3A may be capable of inhibiting the proangiogenic activity of vascular endothelial growth factor (VEGF) [22,42]. Surprisingly, it is suggested that SEMA3A regulates a signaling pathway of its own because SEMA3A promotes vascular permeability, suppresses endothelial cell proliferation, and induces apoptosis when VEGF is not present [23,43,44]. The pro-invasive and pro-metastatic resistance detected upon angiogenesis reduction by the small-molecule tyrosine inhibitor sunitinib in pancreatic neuroendocrine tumors (PNETs) can be overwhelmed by SEMA3A expression [45]. Metastatic PNETs and cervical carcinomas are transformed into benign lesions by re-expressing SEMA3A, since SEMA3A not only enhanced cancer tissue oxygenation and increased the normalization window to inhibit sunitinib-induced activation of epithelial–mesenchymal transition (EMT) and hypoxia-dependent signaling pathways, such as the hypoxia-inducible factor 1-α related pathway, but also prevented tumor hypoxia and restrained cancer dissemination [45]. This logic may lead to the consideration of SEMA3A as a potential therapeutic for the treatment of cancer, specifically in combination with bevacizumab. Bevacizumab blocks the binding of VEGF to its cell surface receptors. This leads to a reduction in the development of vascular structures, as a result, the blood supply to tumor tissues is limited. These effects also reduce tissue interstitial pressure, promote vascular permeability, and support apoptosis of tumor endothelial cells. Increased tumor permeability may increase delivery of chemotherapeutic agents toward the tumor center [46].

Our study had several limitations. First, since our study was retrospective cohort, there could be missing data due to poor registration quality. Therefore, the study was based on a small simple size. Second, we used chronic tonsillitis tissue as a control group because it was nearly impossible to collect normal tonsil tissues. However, inflammation is a well-established hallmark of cancer, therefore tonsillitis tissues may have inflammation-linked molecular abnormalities shared with neoplastic tissues. Generally, comorbid disease in older people is a competing cause of death, which can lead to the development of non-cancer-associated competing mortality. Finally, TNM stage also is a risk factor for poor survival. Lymph node metastasis, which was found in more than 50% of head and neck squamous cell carcinoma patients is associated with poor survival [47]. In addition, tumor size, lymph node involvement and distant metastasis are not only important prognostic factors for survival, but also for recurrence in OPC patients. T classification had a statistically significant influence for patient’s survival [48]. However, in our study, Cox regression did not show any significant associations of those factors perhaps due to a small number of patients. Furthermore, information of potential confounding factors affecting the health of cancer patients such as tobacco or alcohol could not be corrected in our study. Eventually, despite the limitations, this study provides an interesting perspective on the influences of SEMA3A on the clinical parameters of HPV status and its link with neoangiogenesis.

## 5. Conclusions

In our study, we found that there was no relationship between SEMA3A and HPV. In the past few decades, with continued progress in the field of vaccine development to reduce the incidence of cervical cancer, a promising progress for the control of HPV-associated malignancies has been achieved. For the biological similarities between the epithelium of the cervix and the oral cavity, people in general should pay more attention to the role of HPV in OPC. We also found that the loss of SEMA3A expression in OPC and low expression of SEMA3A correlated with poor outcome in patients and high MVD in OPC samples. Taken together, our results demonstrate that SEMA3A serves as a tumor suppressor of OPC tumorigenesis and may become a new target for the treatment of OPC along with therapeutic HPV vaccine.

## Figures and Tables

**Figure 1 microorganisms-08-01286-f001:**
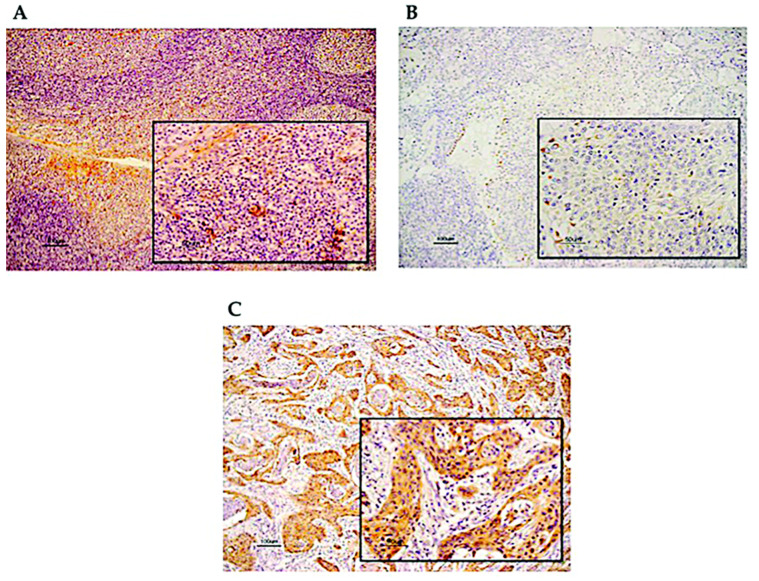
(**A**). SEMA3A in chronic tonsillitis tissue (magnification × 100, scale bar shows 100 µm, lower right rectangle × 400, scale bar shows 50 um). (**B**,**C**). SEMA3A was detected primarily in the cytoplasm of tumor cells (magnification × 100, scale bar shows 100 µm; lower right rectangle × 400, scale bar shows 50 µm) (**B**). Low level of SEMA3A expression (**C**). High level of SEMA3A expression.

**Figure 2 microorganisms-08-01286-f002:**
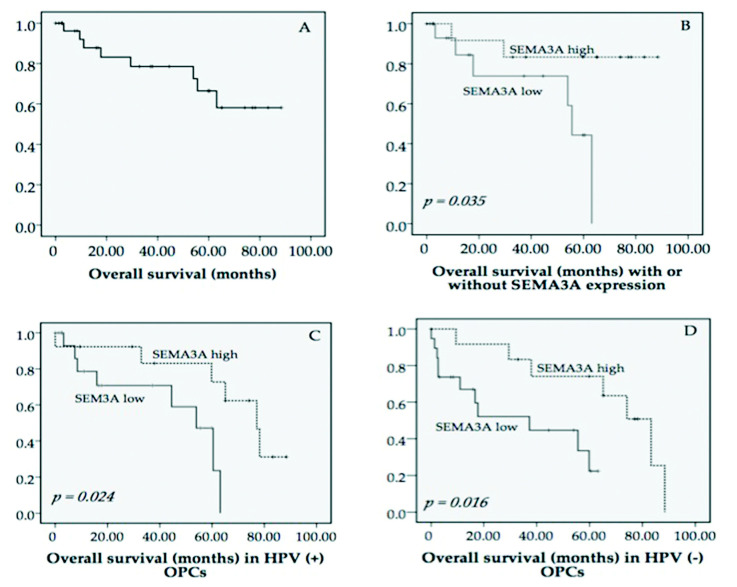
Comparison of different Kaplan–Meier curves for overall survival for patients grouped by immunohistochemistry levels of SEMA3A. (**A**) Kaplan–Meier curves for overall survival (OS) of the 32 OPC patients. (**B**) Kaplan–Meier curves for OS for OPC patients with low and high levels of SEMA3A expression (*p* = 0.035). (**C**) Kaplan–Meier curves for OS in patients who were HPV (+) with low and high levels of SEMA3A expression (*p* = 0.024). (**D**). Kaplan–Meier curves for OS in patients who were HPV (−) with low and high levels of SEMA3A expression (*p* = 0.016).

**Figure 3 microorganisms-08-01286-f003:**
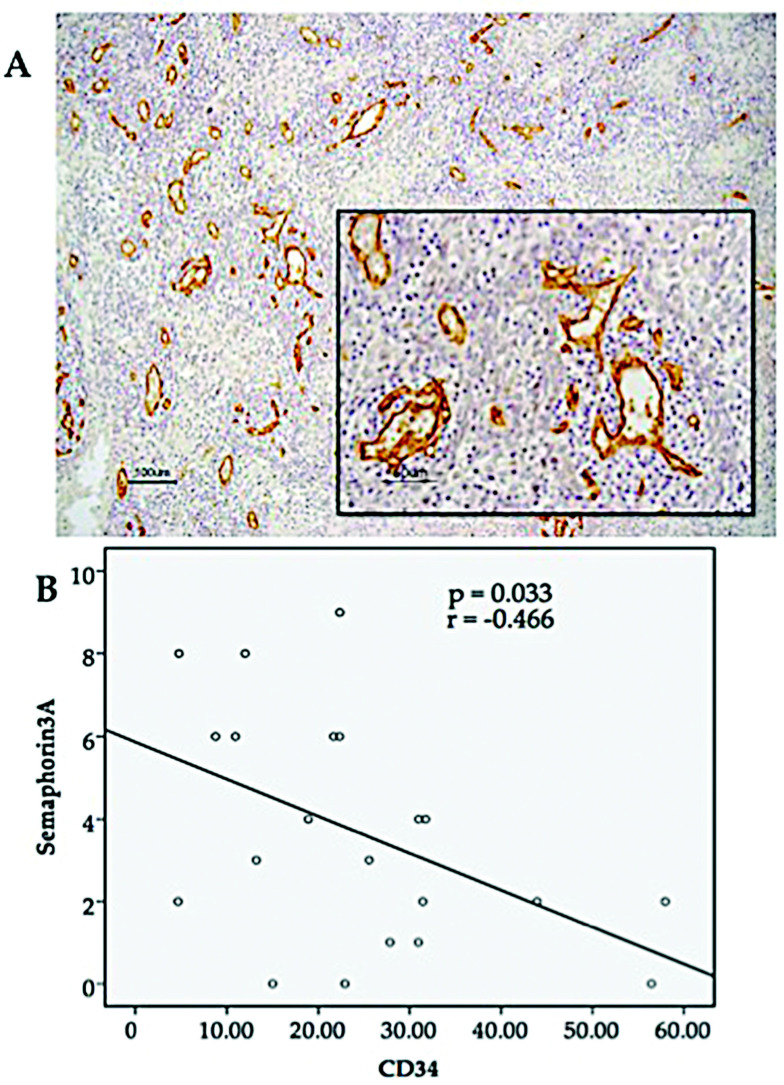
(**A**) Immunohistochemical detection of CD34 protein expression in OPC tissues (magnification × 100, scale bar shows 100 µm; lower right rectangle × 400, scale bar shows 50 µm). (**B**) Relationship between SEMA3A expression and MVD. A Spearman correlation test was performed between SEMA3A expression and CD34 (microvessel density). There was a significant negative correlation between SEMA3A and CD34 (*p* = 0.033, *r* = −0.466).

**Table 1 microorganisms-08-01286-t001:** SEMA3A expression in chronic tonsillitis and OPC tissues (No., number of patients; L, low-expression cohort; H, high-expression cohort; *p*-value < 0.05 was considered statistically significant).

	No.	SEMA3A		
		Low	High	*p*
Tumor	32	19	13	0.005
Chronic tonsillitis	17	3	14	

**Table 2 microorganisms-08-01286-t002:** Relationship between SEMA3A expression in the tumor and clinical characteristics using chi-square test. (L, low-expression cohort; H, high-expression cohort; *p*-value < 0.05 was considered statistically significant).

Characteristic		SEMA3A
		Low	High	*p*
Sex				0.314
Male	25	16	9	
Female	7	3	4	
Age (years)				0.683
≤50	4	2	2	
>50	28	17	11	
Subsite				0.722
Lateral (tonsil and pillars)	21	12	9	
Frontal (base of tongue)Upper (soft palate, uvula)Posterior	11	7	4	
TNM stage				0.654
I-II	14	9	5	
III-IV	16	9	7	
Tumor stage				0.757
T1-2 (early)	19	11	8	
T3-4 (advanced)	11	7	4	
Lymph node metastasis				0.543
N0 (negative)	12	8	4	
N1-3 (positive)	18	10	8	
HPV				0.821
Positive	14	8	6	
Negative	18	11	7	
Smoking				0.515
Never	12	8	4	
Past and present	20	11	9	
Alcohol				0.719
Never	16	10	6	
Past and present	16	9	7	
Histological gradePoorly	11	4	7	0.159
ModeratelyWell	147	105	42

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
