# Peer review of "Influences of Semaphorin 3A Expression on Clinicopathological Features, Human Papillomavirus Status, and Prognosis in Oropharyngeal Carcinoma"

_microorganisms, 2020, doi:10.3390/microorganisms8091286_

Round 1

Reviewer 1 Report

This manuscript provides an interesting and original contribution to the literature. Research into biomarkers with potential prognostic and clinicopathological value in head and neck cancers is a continually developing field of research. The following changes are proposed to improve the article:

- A more explicit and real paragraph showing oropharyngeal cancer epidemiology (number of cases and deaths per year) should be written, using updated references (e.g, GLOBOCAN last report: Bray F, Ferlay J, Soerjomataram I, Siegel RL, Torre LA, Jemal A. Global cancer statistics 2018: GLOBOCAN estimates of incidence and mortality worldwide for 36 cancers in 185 countries. CA Cancer J Clin. 2018;68(6):394-424. doi:10.3322/caac.21492).

- A better elaborated objectives paragraph should be reported. More detailed aims of the study would be advisable (e.g., PECO(S,T) format. In addition, in a concise manner, the authors could include their pre-specified hypotheses.

- The study was apporved by the Ethics Committee of Kanazawa University, but the IRB code is missing.

- The study design was not reported. It seems a retrospective cohort. Authors should confirm it in methodology section and probably discuss their study design limitations in a limitations paragraph.

- The study is based on a very small simple size (32 carcinomas), very probably underpowered to detect significant differences between variables and higher effect sizes (in terms of odds and hazard ratios). This is probably the most important limitations of the present study. It should be commented in the new limitations paragraph in the discussion section.

- The control group tissues derives from patients with tonsillitis. Inflammation is a well-established hallmark of cancer, therefore this tissue may have inflammation-linked molecular abnormalities shared with neoplastic tissue. It should also be argued that it is not the ideal "normal" tissue to be used as a control group.

- The clinicopathological variables recorded and analyzed were all appropiate (and very comprehensive), I only miss histological grade. If possible, it should be included and also statistically analyzed.

- Only overall survival was analyzed as survival variable. If possible, disease-specific survival and recurrence variables (e.g., disease free survival) should be included in the study and analyzed.

- In relation with the immunohistochemical technique, specifically to the antibody used, the manufacturer and dilution were reported, but it is also very important to report the exact clone used, and its nature (mono or polyclonal).

- Some important issues should be better explained and added to the statistical analysis: Table 2 should include a footnote clarifying when chi-square/Fisher exact test were used. Transparency is important to corroborate the appropiate use of both tests.

- Overall survival (and if possible added DSS, DFS, etc) was analyzed using Kaplan-meier curves and probably log rank method (not reported). This variable should be statistically treated as a time-to-event variable, estimating hazard ratios for a better interpretation of results. This is a key point when a prognostic biomarker is under analysis in cancer research (probably the second most important limitations of this study). Nevertheless, it should not be included in the limitations paragraph, I strongly suggest the authors improve this analysis, reporting hazard ratios for time-to-event variables.

- Finally, nothing was discussed about potentially confounding variables (e.g., tobacco, alcohol, age, clinical stage, etc), the tipically Achiles heel of observational studies. It should also be discussed and if posible adjusted in a multivariable model.

Author Response

Reviewer 1

Comments and Suggestions for Authors

This manuscript provides an interesting and original contribution to the literature. Research into biomarkers with potential prognostic and clinicopathological value in head and neck cancers is a continually developing field of research. The following changes are proposed to improve the article:

  1. A more explicit and real paragraph showing oropharyngeal cancer epidemiology (number of cases and deaths per year) should be written, using updated references (e.g, GLOBOCAN last report: Bray F, Ferlay J, Soerjomataram I, Siegel RL, Torre LA, Jemal A. Global cancer statistics 2018: GLOBOCAN estimates of incidence and mortality worldwide for 36 cancers in 185 countries. CA Cancer J Clin. 2018;68(6):394-424. doi:10.3322/caac.21492).

Thank you for your suggestion, I have written down the information accordingly in Introduction section.

  1. A better elaborated objectives paragraph should be reported. More detailed aims of the study would be advisable (e.g., PECO(S,T) format. In addition, in a concise manner, the authors could include their pre-specified hypotheses.

I have added the detailed aims of the study in Introduction, line 80-82.

  1. The study was apporved by the Ethics Committee of Kanazawa University, but the IRB code is missing.

I have added the IRB code in Materials and method section.

4.The study design was not reported. It seems a retrospective cohort. Authors should confirm it in methodology section and probably discuss their study design limitations in a limitations paragraph.

Thank you. This study is a retrospective study. I have added this information in Materials and method section.

  1. The study is based on a very small simple size (32 carcinomas), very probably underpowered to detect significant differences between variables and higher effect sizes (in terms of odds and hazard ratios). This is probably the most important limitations of the present study. It should be commented in the new limitations paragraph in the discussion section.

I have added new paragraph in Dicussion section, line 378-380.

  1. The control group tissues derives from patients with tonsillitis. Inflammation is a well-established hallmark of cancer, therefore this tissue may have inflammation-linked molecular abnormalities shared with neoplastic tissue. It should also be argued that it is not the ideal "normal" tissue to be used as a control group.

Thank you for your suggestion, I have changed all description “normal tonsil” into “chronic tonsillitis tissues” and added new paragraph in Discussion section, line 380-383.

  1. The clinicopathological variables recorded and analyzed were all appropiate (and very comprehensive), I only miss histological grade. If possible, it should be included and also statistically analyzed.

I have added the information for histolocal grade in modified Table 2.

  1. Only overall survival was analyzed as survival variable. If possible, disease-specific survival and recurrence variables (e.g., disease free survival) should be included in the study and analyzed.

DFS have been analyzed following your comments, in Supplemental Figure S1.

  1. In relation with the immunohistochemical technique, specifically to the antibody used, the manufacturer and dilution were reported, but it is also very important to report the exact clone used, and its nature (mono or polyclonal).

I have added the informations in Materials and method section.

  1. Some important issues should be better explained and added to the statistical analysis: Table 2 should include a footnote clarifying when chi-square/Fisher exact test were used. Transparency is important to corroborate the appropiate use of both tests.

I have modified the footnote of Table 2.

  1. Overall survival (and if possible added DSS, DFS, etc) was analyzed using Kaplan-meier curves and probably log rank method (not reported). This variable should be statistically treated as a time-to-event variable, estimating hazard ratios for a better interpretation of results. This is a key point when a prognostic biomarker is under analysis in cancer research (probably the second most important limitations of this study). Nevertheless, it should not be included in the limitations paragraph, I strongly suggest the authors improve this analysis, reporting hazard ratios for time-to-event variables.

Thank you for your suggestion, we have analyzed hazard ratios with OS and DFS included in Supplementary materials Supplemental Table 1 and 2. However, we could not get significant results with Cox regression maybe due to a small number of patients (that you have pointed out). In future, we plan to perform with more patients. In addition, I didn’t include DSS because the results were similar to OS.

  1. Finally, nothing was discussed about potentially confounding variables (e.g., tobacco, alcohol, age, clinical stage, etc), the tipically Achiles heel of observational studies. It should also be discussed and if posible adjusted in a multivariable model.

I have added new paragraph in Discussion section, line 383-393.

Reviewer 2 Report

The article ''Influences of semaphorin 3A expression on clinicopathological features, human papillomavirus status, and prognosis in oropharyngeal carcinoma" is a well-written article by Thanh Hai Pham et al. The article deals with the influence of Semaphorin 3A expression on the clinical parameters of the HPV status and its link with the neoangiogenesis and CD34 expression. The authors results strongly suggest that SEMA3A expression was negatively correlated with CD34 expression which is quite interesting and using the patient samples would add to the significance of the conclusions. Overall the results and analysis provided by authors support the conclusion and my suggestion is the article would be a good interest to the journal readers.

Author Response

Reviewer 2

Comments and Suggestions for Authors

The article ''Influences of semaphorin 3A expression on clinicopathological features, human papillomavirus status, and prognosis in oropharyngeal carcinoma" is a well-written article by Thanh Hai Pham et al. The article deals with the influence of Semaphorin 3A expression on the clinical parameters of the HPV status and its link with the neoangiogenesis and CD34 expression. The authors results strongly suggest that SEMA3A expression was negatively correlated with CD34 expression which is quite interesting and using the patient samples would add to the significance of the conclusions. Overall the results and analysis provided by authors support the conclusion and my suggestion is the article would be a good interest to the journal readers.

Dear Reviewer, we appreciate you for your precious time in reviewing our paper and providing valuable comments.

Round 2

Reviewer 1 Report

All the changes proposed have been taken into account by the authors.